# Proposed Mechanism for the Antitrypanosomal Activity of Quercetin and Myricetin Isolated from *Hypericum afrum* Lam.: Phytochemistry, In Vitro Testing and Modeling Studies

**DOI:** 10.3390/molecules26041009

**Published:** 2021-02-14

**Authors:** Farida Larit, Khaled M. Elokely, Manal A. Nael, Samira Benyahia, Francisco León, Stephen J. Cutler, Mohammed M. Ghoneim

**Affiliations:** 1Département de Chimie, Faculté des Sciences Exactes, Université des Frères Mentouri Constantine 1, Constantine 25000, Algeria; 2Department of BioMolecular Sciences, Division of Medicinal Chemistry, University of Mississippi, University, MS 38677, USA; jleon@mailbox.sc.edu (F.L.); sjcutler@cop.sc.edu (S.J.C.); 3Department of Chemistry, Institute for Computational Molecular Science, Temple University, Philadelphia, PA 19122, USA; kelokely@temple.edu (K.M.E.); mnael@pharm.tanta.edu.eg (M.A.N.); 4Department of Pharmaceutical Chemistry, Faculty of Pharmacy, Tanta University, Tanta 31527, Egypt; 5Laboratoire de Synthèse Organique, Modélisation et Optimisation des Procèdes (LOMOP), Université Badji Mokhtar, Annaba 23000, Algeria; samira.benyahia13@gmail.com; 6Department of Drug Discovery and Biomedical Sciences, College of Pharmacy, University of South Carolina, Columbia, SC 29208, USA; 7Department of Pharmacy Practice, College of Pharmacy, AlMaarefa University, Ad Diriyah, Riyadh 13713, Saudi Arabia; 8Department of Pharmacognosy, Faculty of Pharmacy, Al-Azhar University, Cairo 11371, Egypt

**Keywords:** antitrypanosomal, antileishmanial, hexokinase (TbHK1), docking, flavonoids, *Hypericum afrum*

## Abstract

The in vitro activity of *L. donovani* (promastigotes, axenic amastigotes and intracellular amastigotes in THP1 cells) and *T. brucei*, from the fractions obtained from the hydroalcoholic extract of the aerial part of *Hypericum afrum* and the isolated compounds, has been evaluated. The chloroform, ethyl acetate and *n*-butanol extracts showed significant antitrypanosomal activity towards *T. brucei*, with IC_50_ values of 12.35, 13.53 and 12.93 µg/mL and with IC_90_ values of 14.94, 19.31 and 18.67 µg/mL, respectively. The phytochemical investigation of the fractions led to the isolation and identification of quercetin (**1**), myricitrin (**2**), biapigenin (**3**), myricetin (**4**), hyperoside (**5**), myricetin-3-*O*-β-d-galactopyranoside (**6**) and myricetin-3’-*O*-β-d-glucopyranoside (**7**). Myricetin-3’-*O*-β-d-glucopyranoside (**7**) has been isolated for the first time from this genus. The chemical structures were elucidated by using comprehensive one- and two-dimensional nuclear magnetic resonance (1D and 2D NMR) spectroscopic data, as well as high-resolution electrospray ionization mass spectrometry (HR-ESI–MS). These compounds have also been evaluated for their antiprotozoal activity. Quercetin (**1**) and myricetin (**4**) showed noteworthy activity against *T. brucei*, with IC_50_ and IC_90_ values of 7.52 and 5.71 µM, and 9.76 and 7.97 µM, respectively. The *T. brucei* hexokinase (TbHK1) enzyme was further explored as a potential target of quercetin and myricetin, using molecular modeling studies. This proposed mechanism assists in the exploration of new candidates for novel antitrypanosomal drugs.

## 1. Introduction

Trypanosoma and Leishmania parasites are the etiological agents for the Trypanosomiasis and leishmaniasis diseases that affect millions of people worldwide [1]. The sand-fly is the vector for *Leishmania donovani*, which is the etiological agent of visceral leishmaniasis in humans, while the tsetse fly transmits trypanosomes of *Trypanosoma brucei*, which is the causative agent of human African trypanosomiasis (HAT), also known as sleeping sickness. According to the World Health Organization’s (WHO) statistics, there are 12 million people currently affected by leishmaniasis in 88 countries, and about 350 million people are at risk. Approximately 500,000 people in sub-Saharan Africa are infected annually with the African trypanosome (*T. brucei*), leading to thousands of deaths [1]. The socio-economic impact of the morbidity and mortality caused by these protozoan parasites is significant, particularly in tropical and subtropical countries [2,3].

In the last few decades, there has been an increased focus on developing treatments for African trypanosomiasis. Although several antitrypanosomal agents from plants have been characterized, great efforts are still needed to search for more antiparasitic compounds that have been evolutionarily derived from nature.

The bite of a tsetse fly is responsible for the transmission of *T. brucei,* which quickly adapts to the mammalian host and becomes the bloodstream form (BSF). The metabolism of host glucose through glycolysis is essential for the BSF parasite to successfully infect the host. In BSF parasites, the glycolysis of host glucose provides the sole source of ATP production. The transfer of a phosphoryl group from ATP to glucose in glycolysis is catalyzed by hexokinases (HK). *T. brucei* expresses two hexokinase genes encoding *T. brucei* hexokinase 1 (TbHK1) and 2 (TbHK2), enzymes which are essential for the BSF parasite.

The *Hypericum* genus belonging to the Hypericaceae family has been reported to be a prolific source of various secondary metabolites, such as flavonoids, prenylated phloroglucinols, naphthodianthrones, and volatile oils, with a wide range of biological activities, such as antidepressant, antiseptic, wound-healing, antioxidant, antiviral, anti-inflammatory and antidiabetic. Recently, studies on several Hypericum species have reported their antimicrobial activity against a number of bacterial and fungal strains [4,5,6].

The present study investigates the antileishmanial activity against *L. donovani* (promastigotes, axenic amastigotes and intracellular amastigotes in THP1 cells), and the antitrypanosomal activity against *T. brucei*, of the fractions and isolated compounds of *Hypericum afrum* Lam. Quercetin (**1**) and myricetin (**4**) were discovered to have higher activity against *T. brucei.* Further virtual studies were carried out on TbHK1 as a target enzyme for investigating the probable mechanism of quercetin’s and myricetin’s inhibition of parasitic growth. Quercetin and myricetin bind to TbHK1 proximal to the active site, and could be cytotoxic to the African Trypanosoma parasite as a result of TbHK1 inhibition. 

## 2. Results

### 2.1. Chemistry

Compounds **1**–**7** were identified as quercetin (**1**), myricitrin (**2**), biapigenin (**3**), myricetin (**4**), hyperoside (**5**), myricetin-3-*O*-β-d-galactopyranoside (**6**) and myricetin-3’-*O*-β-d-glucopyranoside (**7**) [7,8,9,10,11,12,13]. Compound **7** was isolated from the genus hypericum for the first time. All the compounds were tested in vitro for their antiprotozoal activity. Pentamidine was used as a positive control drug in the antileishmanial assays, while DMFO (α-difluoromethylornithine) was used as the positive control drug in the antitrypanosomal assay, which showed IC_50_ and IC_90_ values of 3.634 and 8.804 μM, respectively.

### 2.2. Antiprotozoal Activity

In this study, the in vitro antiprotozoal activities were evaluated for the *Hypericum afrum* aerial parts’ fractions (CHCl_3_, EtOAc and *n*-butanol) and isolated pure compounds **1**–**7** (Figure 1; see also Appendix A) against *T. brucei*. The fractions and pure compounds were also tested against *L. donovani* promastigotes, axenic amastigotes and intracellular amastigotes in THP1 cells, for the determination of general toxicity using standard experimental procedures.

None of the fractions or pure compounds showed in vitro antileishmanial activities (Table 1 and Table 2). Regarding the antitrypanosomal activity, the chloroform, ethyl acetate and n-butanol fractions of *H. afrum* showed significant inhibitory activity against *T. brucei* culture, with IC_50_ values of 12.35, 13.53 and 12.93 and with IC_90_ values of 14.94, 19.31 and 18.67 µg/mL, respectively. Only compounds **1** and **4** from the ethyl acetate fraction were found to have antitrypanozomal activity. All the samples were simultaneously tested against THP1 cell for the determination of general cytotoxicity.

Compounds **1** and **4** showed potent activity against *T. brucei* at IC_50_ values of 7.52 and 5.71 and IC_90_ values of 9.76 and 7.97 µM, respectively (Table 1).

### 2.3. Molecular Modeling Study

It is worth noting that the bloodstream form of *T. brucei* is reliant on the glycolytic pathway to generate energy using host glucose as a sole source. Due to the limited homology between the host and *T. brucei* glycolytic enzymes, this makes Hexokinase (TbHK1) a good drug target. Inhibitors of TbHK1 are trypanocidal, with low expected side effects.

A BLAST homology search identified Hexokinase KlHxk1 of the *Kluyveromyces lactis* (PDB Accession code: 3O08) as the most suitable template. KlHxk1 was crystalized with a high resolution of 2 Å. The target and template share 31% sequence identity and 51% similar amino acids, and only 3% gaps were detected. Using protein family information from PFAM, 14 amino acids were identified as having strictly conserved aligned positions of the Hexokinase family, with E-values of 1.7 × 10^−60^, including Thr60, Gly84, Gly92, Arg97, Gly157, Phe158, Phe160, Ser161, Pro163, Leu174, Trp177, Lys179, Asp214 and Gly217 (Figure 2). Three major pockets were detected by Fpocket, one of which overlaps with the possible sugar binding site (Figure 3). This site was selected for the docking of the compounds. Myricetin showed a docking score of −8.31 (and IFD score: −743.41), compared to −6.62 (IFD score: −735.48) for quercetin. The two compounds occupied the same pocket with very similar conformation (Figure 4).

Quercetin exhibited hydrogen bonds with Ser161, Arg176, Thr178, Thr215 and Glu269, and π–π interaction with Phe162. The additional hydroxyl group in myricetin did not provide extra important interaction with the surrounding amino acids. On the other hand, some of the interactions were lost, which could be attributed to a slight shifting of the interacting functional groups of myricetin to allow for a proper placement of the additional hydroxyl group inside the binding site. Molecular dynamics (MD) simulations were conducted for 50 ns to study the stability and strength of ligand binding. The root means square deviation (RMSD) is used as an indicator for interaction stability.
$$RMSD_{x}=\sqrt{\frac{1}{N}\overset{N}{\underset{i=1}{\sum }}{\left(r_{i}^{\prime }\left(t_{x}\right)-r_{i}\left(t_{ref}\right)\right)}^{2}}$$
*N* is the number of atoms in the system,*t_ref_* is the reference time,*r′_i_* is the position of the selected atoms (*i*) in frame *x*, *t_x_* is the record time of frame *x*.

Myricetin showed a stable pose in the binding pocket over the course of the MD time. It showed an RMSD value of ~3.0 Å, which is comparable to the 2.4 Å of the protein backbone, while quercetin showed a high RMSD value of ~15.0 Å compared to that of the protein backbone (2.9 Å). The binding mode of quercetin required the first 10 ns to adopt the most stable pose in the binding pocket (Figure 5). The reference interaction site model (RISM) solvation approach was used to investigate the location and stability of water molecules (Figure 6 and Figure 7).

Possible water molecule locations were computed and thermodynamically analyzed. The absolute protein–ligand binding free energies were then calculated using the water swap method to rank quercetin and myricetin, by relying on the fact that protein–ligand binding is a competition between ligand and water for binding to the pocket (Figure 8). Because this approach allows for the decomposition of ΔG on a per-residue basis, it permits the identification of which amino acid residues favor binding to ligand or water (Figure 9). The free energy of binding is calculated in kcal/mol for each ligand with different methods, including Bennett, thermodynamic integration (TI), free energy perturbation (FEP) and TI Quadrature. The results in kcal/mol for myricetin are as follows: Bennett, −15.34; TI, −14.15; FEP, −14.14; and TI Quadrature, −14.23. Those of quercetin are Bennett, −9.92; TI, −9.28; FEP, –8.84; and TI Quadrature, −9.6. Then, a consensus-calculated free energy of binding was defined from a weighted arithmetic mean of the Bennett (50%), TI (30%), and FEP (20%) free energy estimators. The consensus is −14.74 for myricetin and −9.51 for quercetin.

## 3. Discussion

Flavonoids isolated from different natural resources were reported to exhibit moderate to high in vitro antitrypanosomal activity against *T. brucei*. Our results showed that the chloroform, ethyl acetate and *n*-butanol fractions of *H. afrum* exhibited potent antitrypanosomal activity against *T. brucei*, with IC_50_ values of 12.35, 13.53 and 12.93 and with IC_90_ values of 14.94, 19.31 and 18.67 µg/mL, respectively. Quercetin and myricetin were isolated from the EtOAc fraction, and showed good activity towards *T. brucei*, with IC_50_ values of 7.52 and 5.71, and IC_90_ values of 9.76 and 7.97 µM, respectively. A future examination of the chloroform and *n*-butanol fractions might yield bioactive molecules.

Quercetin and structurally related flavonoids possess several biological and pharmacological activities. Quercetin and myricetin are dietary flavonoids with promising activities, including antioxidant, anti-inflammatory, cardiovascular, and others [14,15]. The mechanism of action for quercetin and myricetin as antiparasitics has been postulated as follows: these flavonoids can disrupt the mitochondrial function on the parasites, and most likely inhibit different enzymes, including shock proteins, topoisomerases and kinases, among others. They could also show indirect activity through the induction of microbicidal responses, for example, the production of various cytokines and the production of nitric oxide [16]. Hexokinase (TbHK1) enzymes have been shown to be promising targets on *T. brucei* [17,18]. Quercetin and myricetin were identified as inhibitors of TbHK1, and showed IC_50_ values of 4.1 and 48.9 µM, respectively [16]. These results support the idea of considering TbHK1 as a target for antiparasitic activities. These results were further confirmed by our computational studies.

Quercetin has been reported as a potent antileishmanial agent; however, in our study, we did not find it [19].

## 4. Material and Methods

### 4.1. General Experimental Procedures

Melting points were determined on an Opti-Melt automated melting point system (Stanford Research Systems), and were uncorrected. IR spectra were recorded using an Agilent model Cary 630 FT-IR. Optical rotations were recorded using a Rudolph Research Analytical Autopol V Polarimeter. UV was obtained using a Perkin–Elmer Lambda 3B UV/vis spectrophotometer. ^1^H- and ^13^C-NMR spectra were obtained on Bruker model AMX 500 and 400 NMR spectrometers with standard pulse sequences, operating at 500 and 400 MHz in ^1^H and 125 and 100 MHz in ^13^C. The coupling constants were recorded in Hertz (Hz). Standard pulse sequences were used for COSY, HMQC, HMBC and DEPT. All spectra were run at 25 °C. High-resolution mass spectra (HRMS) were measured on a Micromass Q-Tof Micro mass spectrometer with a lock spray source. Column chromatography was carried out on silica gel (70–230 mesh, Merck, Darmstadt, Germany), C18 SPE (500 mg Bed, Thermo scientific, Waltham, MA, USA), Diaion HP-20 (Sorbetch technologies, Norcross, GA, USA), MN-polyamide-SC-6 (Sigma Aldrich, St Louis, MO, USA) and Sephadex LH-20 (Sigma Aldrich, St Louis, MO, USA). TLC (silica gel 60 F254, EmD Millipore Sigma, Darmstadt, Germany) was used to monitor the fractions from column chromatography. Preparative TLC was carried out on silica gel 60 PF254 + 366 plates (20 cm × 20 cm, 1 mm thick). Visualization of the TLC plates was achieved with a UV lamp (λ = 254 and 365 nm) and an anisaldehyde/acid spray reagent (MeOH-acetic acid-anisaldehyde-sulfuric acid, 85:9:1:5). The absolute configurations of the sugar moieties of all tested flavonoid glycosides have been identified by the UPLC-UV/MS method [20].

### 4.2. Plant Material

The aerial parts of *Hypericum afrum* (Lam.) were collected from the El Kala region, El Tarf, in northeastern Algeria, in July 2011. A voucher specimen (UM-10012014) was deposited in the culture collection of the Department of BioMolecular Sciences, University of Mississippi.

### 4.3. Extraction and Isolation

Extraction and isolation. Dried powdered aerial parts (1000 g) of *H. afrum* were macerated at room temperature with EtOH–H_2_O (80:20, *v/v*) over 24 h, three times. The filtered solvents were combined and evaporated under vacuum at a temperature of 40 °C to give a residue (30 g). The obtained extract was suspended in water (800 mL) and successively partitioned with CHCl_3_, EtOAc and *n*-butanol, yielding 500 mg (CHCl_3_), 7g (EtOAc) and 12g (*n*-butanol) fractions, respectively.

The EtOAc fraction (7 g) was chromatographed on a silica gel column and eluted with a CH_2_Cl_2_–MeOH solvent system of increasing polarity to yield 10 subfractions (E-1–10) according to their TLC behavior. The subfraction E-3 (115mg) was further chromatographed on the column of a Sephadex LH-20 with CH_2_Cl_2_–MeOH (1:1) as the eluent yielding compound **1** (16 mg) as a yellow precipitate. A yellow precipitate was obtained from the subfraction E-7 (40% MeOH in CH_2_Cl_2_). The solid was combined and subjected to a column of Sephadex LH-20 eluted with methanol to furnish compound **2** (7 mg). The subfraction E-4 (10% MeOH) (423 mg) was chromatographed on Sephadex LH-20 eluted with methanol to furnish compound **3** (10 mg). Subfraction E-5 (125 mg) was chromatographed on a Sephadex LH-20 column with Methanol and further purified by preparative TLC eluted with CHCl_3_–MeOH (10:1) to afford compound **4** (15 mg).

The *n*-butanol fraction (12 g) was subjected to column chromatography over Diaion HP-20 to afford three subfractions: A (H_2_O 100%), B (50% Me0H in water) and C (100% MeOH). Subfraction C (8 g) was subjected to polyamide column chromatography to give 13 subfractions (C-I to C-XIII). Then, 5 g of the subfraction C-X (100% methanol) was chromatographed on MN-polyamide-SC-6 CC (250 g), eluted with water to equilibrate, then with gradient–decreased polarities with a water–methanol system to yield 8 subfractions (C-X-1 to C-X-8). Subfraction C-X-4 (50 mg) was further subjected to a column of Sephadex LH-20 (20 g) eluted with MeOH-CH_2_Cl_2_ (1:1) to afford compound **5** (4.0 mg). C-X-5 (70 mg) was subjected to a column of Sephadex LH-20 (30 g) eluted with MeOH-CH_2_Cl_2_ (1:1) to afford compounds **6** (5.0 mg) and **7** (10.6 mg).

### 4.4. In Vitro Antileishmanial and Antitrypanosomal Assays

The in vitro antileishmanial and antitrypanosomal assays were performed on cell cultures of *L. donovani* promastigotes, axenic amastigotes, THP1-amastigotes, and *Trypanosoma brucei* trypomastigotes, by Alamar Blue assays [21]. The conditions for seeding the THP1 cells, the exposure to the test samples and the evaluation of cytotoxicity were the same as described in the parasite rescue and transformation assay [22]. IC_50_ and IC_90_ values were computed from the dose–response curves using XLfit software. DFMO (difluoromethylornithine) was used as the positive control. The antiprotozoal activities of the *H. afrum* fractions and isolated compounds were evaluated in vitro against *L. donovani* promastigotes, axenic amastigotes and intracellular amastigotes in THP1 cells. The fractions and some isolated compounds were also evaluated against *T. brucei* trypomastigote forms. All the fractions and compounds were simultaneously tested against THP1 cell for the determination of general cytotoxicity [22].

### 4.5. Methods: Molecular Modeling

The amino acid sequence of TbHK1 was downloaded from the Uniprot database (www.uniprot.org)(accessed on 1 December 2019). Prime was used to build the 3D model of the target [23,24,25]. To find the appropriate template, a BLAST homology search was initiated with default options. The globally conserved residues were also identified through the standard Prime search algorithm. The alignment between the target sequence and the template was calculated to define the complementary secondary structure information using the Prime STA option. An all-atom model was constructed using the knowledge-based method. Hydrogen bond assignment and restrained minimization options were used to fix and relax the generated model. Loop refinement was performed for the model to optimize the loops’ conformation. Fpocket [26] was then used to detect the ligand binding sites in the model. We used Glide for the molecular docking step. The receptor grid was prepared by selecting the amino acids defining the pocket detected by Fpocket. To allow for more space for the ligands in the binding pocket, scaling of the van der Waals radii by a factor of 0.8 was employed to soften the potential of the nonpolar parts of the receptor. The prepared ligands were docked using Glide SP docking precision [27,28,29,30,31]. The best-docked pose was saved for each ligand. Then, these poses were used for the molecular dynamics simulation [32,33], 3D-RISM [34,35,36,37] and WaterSwap [38,39] steps. The two complexes were solvated in an orthorhombic box using a TIP4P water solvation model in Desmond System Builder. Appropriate numbers of sodium ions were added to the system to neutralize the net charge on the protein. The OPLS3 force field was selected for the simulation run. After a series of energy minimizations and short simulations, the NPT ensemble was set for the MD production step. Intervals of 25 ps were used to save the coordinates and the MD simulations were set for 50 ns. The protein–ligand complex was prepared as recommended for the 3D-RISM calculation. The grid was defined for the whole protein and the calculation was performed using the XED force field. We used a grid spacing of 0.3 Å with a grid external width of 14.0 Å. The convergence tolerance was set to 10^−8^ and the total formal charge was neutralized with counter ions. A WaterSwap calculation was conducted for the ligands. The normal calculation method was chosen. The charge method was set to Gasteiger, and the solvation box buffer dimension was defined as 10.0 Å. The system was equilibrated for 100 ps using OpenMM.

## 5. Conclusions

The in vitro antitrypanosomal evaluation of the *Hypericum afrum* Lam. species against *T. brucei* revealed that the CHCl_3_, EtOAc and BuOH fractions possess potent antitrypanosomal activity. Furthermore, the fractionation of these fractions led to the isolation and characterization of quercetin and myricetin as the potent components. Quercetin (**1**) and myricetin (**4**) showed good antitrypanosomal activity towards *T. brucei*, with IC_50_ and IC_90_ values of 7.52 and 5.71, and 9.76 and 7.97 µM, respectively.

The present study reports for the first time the antiprotozoal properties of the Algerian species *Hypericum afrum* Lam., contributing to the phytochemical and pharmacological knowledge of this species. Moreover, this species could be a source of new antitrypanosomal agents.

The mechanism of antitrypanosomal activity was investigated using molecular docking studies on the potential target enzyme *T. brucei* Hexokinase (TbHK1). Docking studies of molecules (**1**) and (**4**) revealed their strong affinity towards Hexokinase (TbHK1) as a target of *T. brucei*.

## Figures and Tables

**Figure 1 molecules-26-01009-f001:**
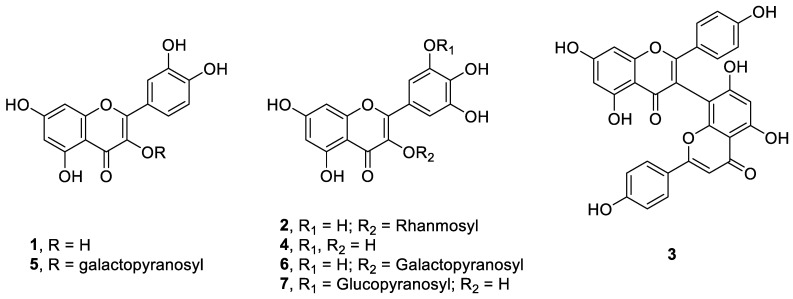
Structure of isolated compounds (**1**–**7**).

**Figure 2 molecules-26-01009-f002:**
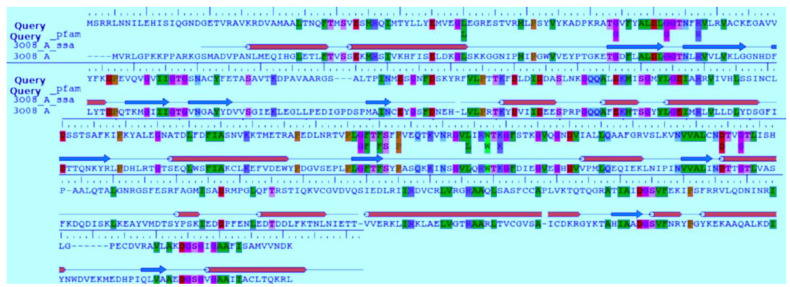
The secondary structure and conservation information of the target. Secondary structure predictions for the target are annotated as Hs (helices) and Es (beta sheets), while the structure of the template is shown as blue arrows for beta sheets and orange cylinders for alpha helices. Globally conserved regions are annotated as a separate row.

**Figure 3 molecules-26-01009-f003:**
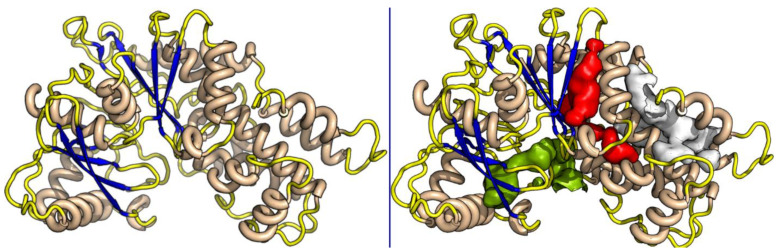
The 3D model of TbHK1 (**left**) and the most prominent pockets for ligand binding (**right**).

**Figure 4 molecules-26-01009-f004:**
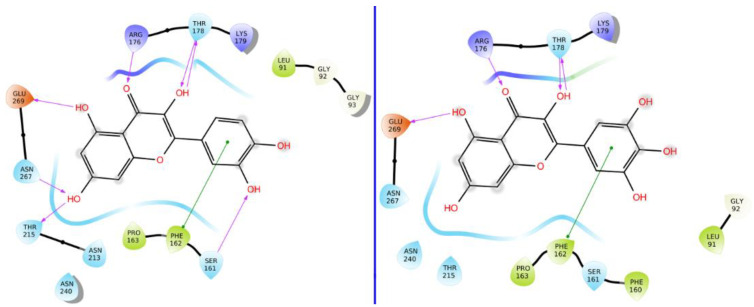
The 2D interaction models of quercetin (**left**) and myricetin (**right**).

**Figure 5 molecules-26-01009-f005:**
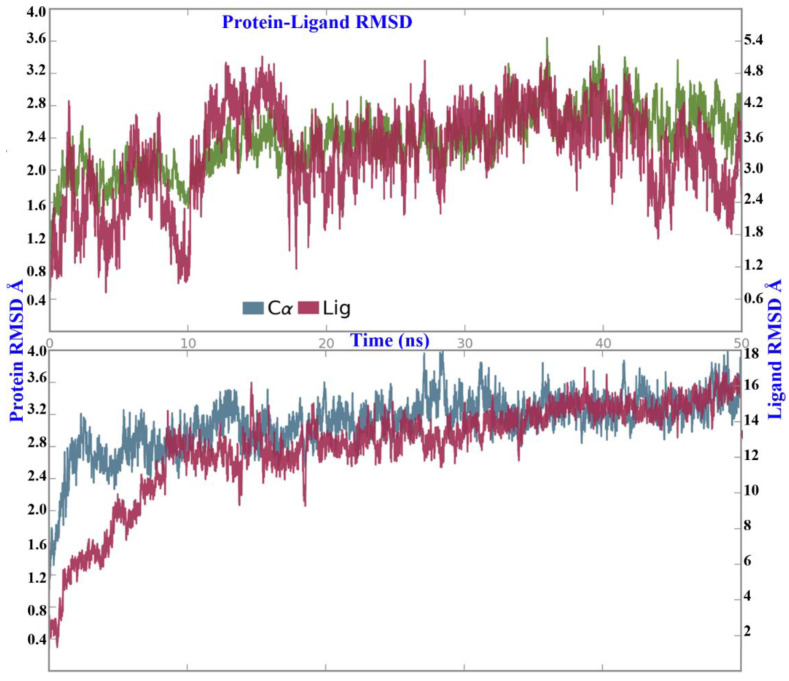
The RMSD of protein and ligand over the MD simulation time.

**Figure 6 molecules-26-01009-f006:**
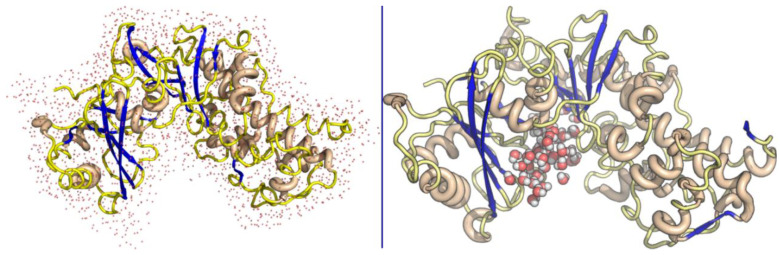
The solvated protein (**left**) and solvated binding pocket before interaction with the ligand (**right**).

**Figure 7 molecules-26-01009-f007:**
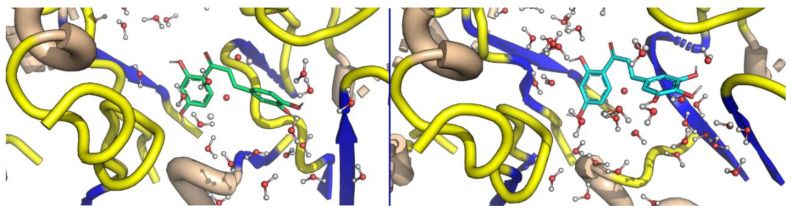
The ligands displace water molecules from the binding pocket.

**Figure 8 molecules-26-01009-f008:**
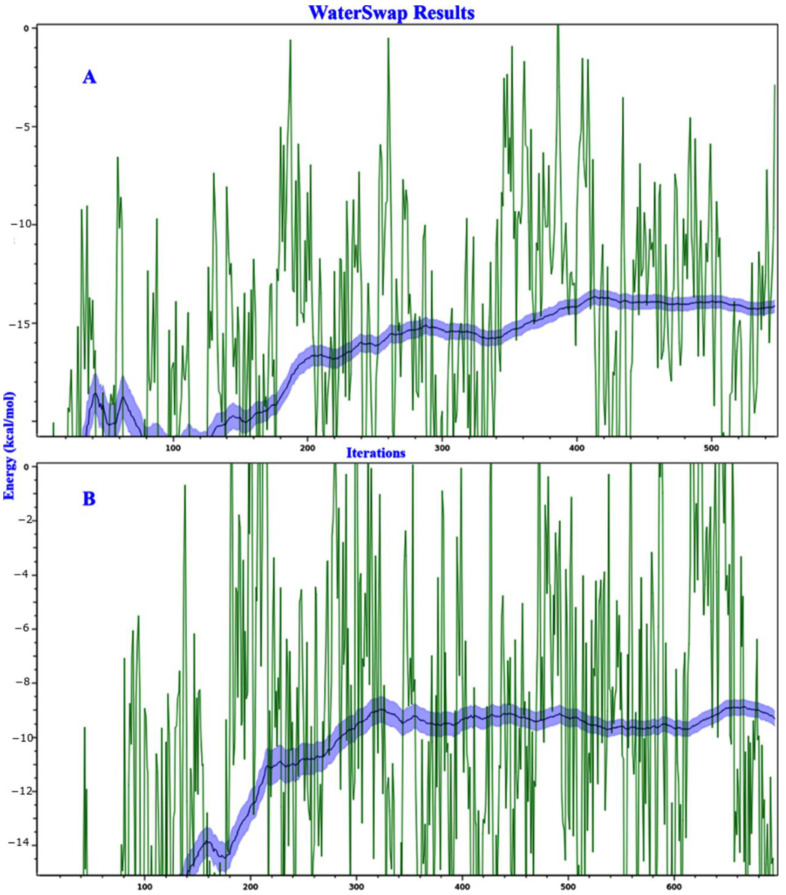
Water Swap results of myricetin (**A**) and quercetin (**B**).

**Figure 9 molecules-26-01009-f009:**
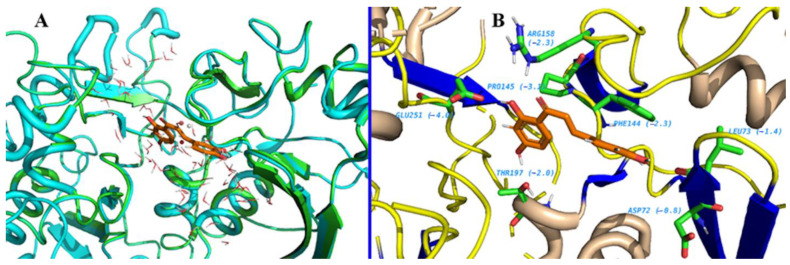
(**A**) The induced conformational changes upon ligand (quercetin) binding. The protein is shown as a cyan cartoon before binding of the ligand, and the conformational changes are shown in green. (**B**) The regions of the pocket that favor ligand chemistry are shown as green sticks. The free energy (ΔG) of each amino acid is shown in parenthesis; the more negative, the better.

**Table 1 molecules-26-01009-t001:** Antiprotozoal activity and cytotoxicity of isolated compounds of *H. afrum.*

Compounds Codes	Compounds Names	*L. Donovani* Promastigote IC_50_ (μM)	*L. Donovani* Promastigote IC_90_ (μM)	*L. Donovani* Amastigote IC_50_ (μM)	*L. Donovani* Amastigote IC_90_ (μM)	*L. Donovani* Amastigote/THP1 IC_50_ (μM)	*L. Donovani* Amastigote/THP1 IC_90_ (μM)	*T. Brucei* IC_50_ (μM)	*T. Brucei* IC_90_ (μM)	THP1 Cytotoxicity IC_50_ (μM)	THP1 Cytotoxicity IC_90_ (μM)
AMB	Amphotericin	0.136	0.215	0.211	0.374	0.188	0.421	NT	NT	>2	>2
PENT	Pentamidine	1.478	2.382	9.581	>10	1.157	5.587	0.001	0.002	>10	>10
DFMO	difluoromethylornithine	NT	NT	NT	NT	NT	NT	3.634	8.804	NT	NT
**1**	Quercetin	>10	>10	>10	>10	>10	>10	7.52	9.76	>10	>10
**2**	Miricitrin	>10	>10	>10	>10	>10	>10	>10	>10	>10	>10
**3**	Biapigenin	>10	>10	>10	>10	>10	>10	>10	>10	>10	>10
**4**	Myricetin	>10	>10	>10	>10	>10	>10	5.71	7.97	>10	>10
**5**	Hyperoside	>10	>10	>10	>10	>10	>10	>10	>10	>10	>10
**6**	Myricetin-3-*O*-β-d-glucopyranoside	>10	>10	>10	>10	>10	>10	>10	>10	>10	>10
**7**	Cannabiscitrin	>10	>10	>10	>10	>10	>10	>10	>10	>10	>10

NT, not tested; IC-50 and IC-90 values are expressed as µM and are the mean ±S.D. of duplicate observations. Tested concentrations range: 0.4–10 ug/mL.

**Table 2 molecules-26-01009-t002:** Antiprotozoal activity and cytotoxicity of *H. afrum* fractions.

Fractions	*L. Donovani* Promastigote IC_50_ (μg/mL)	*L. Donovani* Promastigote IC_90_ (μg/mL)	*L. Donovani* Amastigote IC_50_ (μg/mL)	*L. Donovani* Amastigote IC_90_ (μg/mL)	*L. Donovani* Amastigote/THP IC_50_ (μg/mL)	*L. Donovani* Amastigote/THP IC_90_ (μg/mL)	*T. Brucei* IC_50_ (µg/mL)	*T. Brucei* IC_90_ (µg/mL)	THP1 Cytotoxicity IC_50_ (μg/mL)	THP1 Cytotoxicity IC_90_ (μg/mL)	Test conc. (µg/mL)
CHCl_3_	>20	>20	>20	>20	>20	>20	12.35	14.94	>20	>20	20–0.8
EtOAc	>20	>20	>20	>20	>20	>20	13.53	19.31	>20	>20	20–0.8
BuOH	>20	>20	>20	>20	>20	>20	12.93	18.67	>20	>20	20–0.8
AMB	0.138	0.188	0.304	0.362	0.187	0.264	NT	NT	>2	>2	2.0–0.08
PENT	1.478	2.382	9.581	>10	1.157	5.587	0.001	0.002	>10	>10	0.02
DFMO	NT	NT	NT	NT	NT	NT	3.634	8.804	NT	NT	20–0.8

NT, not tested; IC_50_ and IC_90_ values are expressed as µg/mL.

## Data Availability

Data is available from the authors.

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
