# Peer review of "Proposed Mechanism for the Antitrypanosomal Activity of Quercetin and Myricetin Isolated from Hypericum afrum Lam.: Phytochemistry, In Vitro Testing and Modeling Studies"

_molecules, 2021, doi:10.3390/molecules26041009_

Round 1

Reviewer 1 Report

In general they have a series of errors that they must correct. Furthermore, the discussion is poor regarding several items: mechanism of action, cytotoxicity, activity against plasmodium falciparum.

In the abstract they do not include everything they mention they did, it does not match the title of the manuscript.

The authors mention that they did activity tests against Plasmodium falciparum, however they do not present results about this.

Regarding cytotoxic activity, they had to evaluate higher concentrations, and discuss this part. It is important to know the difference between the antiprotozoal and cytotoxic activity of the tested compounds and fractions.

Errors were marked in yellow, and messages with comments were included within the PDF file.

Author Response

Reviewer 1

We greatly appreciate the constructive comments and suggestions provided by the editor and reviewers, and we have revised our manuscript to address all comments and concerns on a point-by-point basis and are presented in the following pages. With this submission, we are including the revised version using Track Changes Function. We believe that the revision process has resulted in a significantly sound manuscript. We hope that our article will be of interest to the broad readership of Molecules

Reviewer # 1

Comments: “In general, they have a series of errors that they must correct. Furthermore, the discussion is poor regarding several items: mechanism of action, cytotoxicity, activity against plasmodium falciparum. In the abstract they do not include everything they mention they did, it does not match the title of the manuscript. The authors mention that they did activity tests against Plasmodium falciparum, however they do not present results about this. Regarding cytotoxic activity, they had to evaluate higher concentrations, and discuss this part. It is important to know the difference between the antiprotozoal and cytotoxic activity of the tested compounds and fractions. Errors were marked in yellow, and messages with comments were included within the PDF file.”

Response: We thank the reviewer for the constructive comments, suggestions, and corrections. We fixed the errors and revised the manuscript.

Comment: In general, they have a series of errors that they must correct. Furthermore, the discussion is poor regarding several items: mechanism of action, cytotoxicity, activity against plasmodium falciparum.

Response: The activity against plasmodium falciparum has been deleted because it was added by mistake. We extended the discussion about the mechanism.

Comment: In the abstract they do not include everything they mention they did, it does not match the title of the manuscript.

Response: The abstract is rewritten to reflect the manuscript’s title and content.

Comment: The authors mention that they did activity tests against Plasmodium falciparum, however they do not present results about this.

Response: Thank you for the comment. We have corrected the manuscript, and everything about antimalarial activity has been deleted from the text. 

Comment: Regarding cytotoxic activity, they had to evaluate higher concentrations, and discuss this part. It is important to know the difference between the antiprotozoal and cytotoxic activity of the tested compounds and fractions.

Response: We thank the reviewer for the suggestion about the cytotoxicity activity. In fact, we followed the protocols given in PMID: 23299097, and evaluating a higher concentration for THP1 will not give us extra information.

Comment: Errors were marked in yellow, and messages with comments were included within the PDF file.

Response: The manuscript has been revised and all highlighted errors have been taken care of.

Reviewer 2 Report

The antileishmanial and antitrypanosomal activities of many flavonoids, including the only two active compounds of this study, myricetin and quercetin are already reported, the antileishmanial activity even in vivo, in Ref. 20 (this is a wrong citation, there is nothing to find about hexokinases..)

The 13C spectra of compound 1 has a carbon too much (at 49 ppm, what is that ?)

Which solvents were used for NMR ?

Page 3, line 138: the antiprotozoal activity of C. villosus ? why that plant ? must be a mistake..

Line 143, same page: The extract and isolated compounds were also evaluated for their antimalarial activity... i did not found any results... 

Author Response

Reviewer -2

We greatly appreciate the constructive comments and suggestions provided by the editor and reviewers, and we have revised our manuscript to address all comments and concerns on a point-by-point basis and are presented in the following pages. With this submission, we are including the revised version using Track Changes Function. We believe that the revision process has resulted in a significantly sound manuscript. We hope that our article will be of interest to the broad readership of Molecules

Reviewer # 2

Comment: “The antileishmanial and antitrypanosomal activities of many flavonoids, including the only two active compounds of this study, myricetin and quercetin are already reported, the antileishmanial activity even in vivo, in Ref. 20 (this is a wrong citation, there is nothing to find about hexokinases..). The 13C spectra of compound 1 has a carbon too much (at 49 ppm, what is that ?). Which solvents were used for NMR ?. Page 3, line 138: the antiprotozoal activity of C. villosus ? why that plant ? must be a mistake.. Line 143, same page: The extract and isolated compounds were also evaluated for their antimalarial activity... i did not found any results...”

Response: We thank the reviewer for the constructive comments and we adjusted the manuscript to address the comments.

Comment: The antileishmanial and antitrypanosomal activities of many flavonoids, including the only two active compounds of this study, myricetin and quercetin are already reported, the antileishmanial activity even in vivo.

Response: Thank you for this comment and the reference was added to reflect the previous report. In the current manuscript we report the activity of these isolates from H. afrum and further virtually confirming and exploring mechanism of action of quercetin and myricetin against TBHK1 enzyme.

Comment: in Ref. 20 (this is a wrong citation, there is nothing to find about hexokinases...)

Response: The reference was changed.

Comment: The 13C spectra of compound 1 has a carbon too much (at 49 ppm, what is that?)

Response: The NMR was performed in DMSO and the signal at 49 ppm is the signal residual for methanol as an impurity.

Comment: Which solvents were used for NMR?

Response: The solvents that have been used are: DMSO-d6, and Methanol-d4

Comment: Page 3, line 138: the antiprotozoal activity of C. villosus ? why that plant ? must be a mistake..

Response: Thank you for spotting that, it was a mistake. The name of the plant has been corrected in the same phrase.

Comment: Line 143, same page: The extract and isolated compounds were also evaluated for their antimalarial activity... i did not found any results...

Response: Thank you for your comment. We corrected the manuscript and everything about antimalarial activity has been deleted.  

Round 2

Reviewer 1 Report

Errors were marked in messages box within the file.

Author Response

Authors’ Responses to Reviewer Comments

We greatly appreciate comments and suggestions provided by the reviewer, and we have revised our manuscript to address all comments and concerns on a point-by-point basis, we are including the revised version using Track Changes Function.

Reviewer # 1

Comments: Errors were marked in messages box within the file.

Response: We thank the reviewer for his comments and corrections. All errors have been fixed and we are including the revised version using Track Changes Function.